# Neural-Symbolic Entangled Framework for Complex Query Answering

**Zezhong Xu**[1],[*] **Wen Zhang**[1],[*] **Peng Ye**[1], **Hui Chen**[3], **Huajun Chen**[1],[2],[†]

[1]Zhejiang University & AZFT Joint Lab for Knowledge Engine, China
[2]Hangzhou Innovation Center, Zhejiang University, [3]Alibaba Group
{xuzezhong, zhang.wen, yep, huajunsir}@zju.edu.cn,
weidu.ch@alibaba-inc.com

## Abstract

Answering complex queries over knowledge graphs (KG) is an important yet challenging task because of the KG incompleteness issue and cascading errors during reasoning. Recent query embedding (QE) approaches embed the entities and relations in a KG and the first-order logic (FOL) queries into a low dimensional space, answering queries by dense similarity search. However, previous works mainly concentrate on the target answers, ignoring intermediate entities' usefulness, which is essential for relieving the cascading error problem in logical query answering. In addition, these methods are usually designed with their own geometric or distributional embeddings to handle logical operators like union($\vee$), intersection($\wedge$), and negation($\neg$), with the sacrifice of the accuracy of the basic operator – projection, and they could not absorb other embedding methods to their models. In this work, we propose a **Ne**ural and **Sy**mbolic **E**ntangled framework (**ENeSy**) [3]) for complex query answering, which enables the neural and symbolic reasoning to enhance each other to alleviate the cascading error and KG incompleteness. The projection operator in ENeSy could be any embedding method with the capability of link prediction, and the other FOL operators are handled without parameters. With both neural and symbolic reasoning results contained, ENeSy answers queries in ensembles. ENeSy achieves the SOTA performance on several benchmarks, especially in the setting of training model only with the link prediction task.

## 1 Introduction

People built different Knowledge Graphs, such as Freebase [4], YAGO [13], and Wordnet [11], to store complex structured information and knowledge. The facts in KG are usually represented in the form of triplets, e.g., *isCityOf(New York, USA)*. KGs have been widely applied in various intelligent systems such as question answering and natural language understanding. One of the key tasks on KG reasoning is complex query answering which involves answering FOL query with logical operators including existential quantification ($\exists$), conjunction($\wedge$), disjunction($\vee$), and negation($\neg$).

Given a question *"Who won the Turing Award in developing countries?"*, as illustrated in Figure 1, it could be converted to a FOL query, and a computation graph can be generated with the query. Each node in the computation graph represents an entity or an entity set, while each edge represents a logical operation. Answering these queries is challenging since not all the answers could be directly identified by traversing the KG because of the incompleteness of KG. To address this problem, several

---

[*]Equal Contribution.
[†]Corresponding Author.
[3]Source code of ENeSy is available at https://github.com/zjukg/ENeSy.

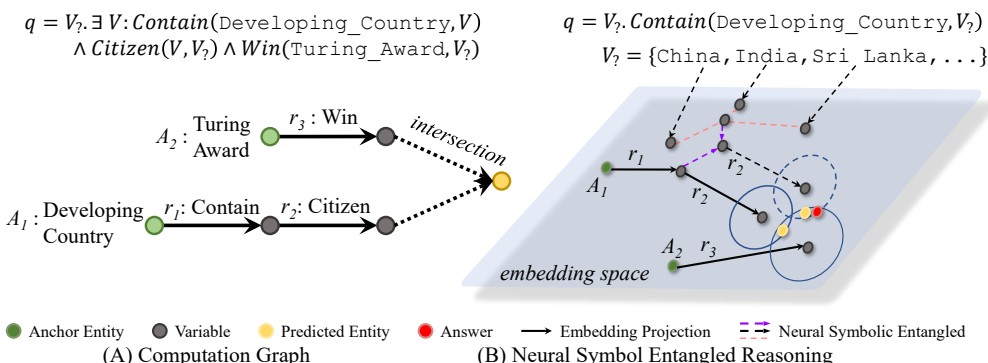

Figure 1: An example of answering a complex graph query by using the ENeSy. (A): FOL query and its computation graph for the question *'Who won the Turing Award in developing countries?'*. (B): ENeSy uses neural and symbolic ways to handle projection separately, and the results are entangled to enhance each other to alleviate the problem of cascading error and incompleteness of KG. The logic operator $\wedge$, $\vee$, and $\neg$ are supported with symbolic reasoning.

QE methods [9, 14, 15, 6] are proposed which encode entities and the query to a low-dimensional embedding space and the logical operators are also parameterized. Following the corresponding computation graph, the query embedding could be computed step by step. The target answers are selected according to the distance between the final embedding and candidate entity embedding.

Although the query embedding methods are effective for solving the incompleteness of KG, several limitations still exist. Firstly, the role of intermediate entities remains largely unexamined which we argue plays a significant role in obtaining the target answers since besides incompleteness, cascading error also influences the accuracy of complex queries. Existing works only pay attention to the target answers, but for multi-hop reasoning, the intermediate entities could not be ignored. Secondly, previous works propose different geometric shapes or distribution embedding to support more logical operators. For instance, Query2Box (Q2B) embed queries to boxes and can handle intersection($\wedge$), while ConE [26] and BetaE [15] embed query with cone embedding and beta distribution to further support negation($\neg$). Although the logical operators are supported with their own embedding shape, the performance of the basic operator, projection, is not satisfying. For example, the accuracy of Q2B on one projection query is worse than TransE [5] according to the Q2B paper [14]. Few studies have investigated generalizing existing embedding models, such as TransE and RotatE which achieve good performance on link prediction, to the complex query.

In this paper, we propose a neural symbolic entangled model, named ENeSy, which enables embedding and symbolic reasoning to enhance each other. The embedding results influence the symbolic results to solve the KG incompleteness while symbolic results could revise embedding results by alleviating the problem of cascading error in multi-hop reasoning. Our approach has the following important advantages: (1) ENeSy can generalize KG embedding methods to answering complex queries, and it could not only use the classic KGE algorithm like TransE, but also the projection operator of existing query embedding methods could be employed. (2) The logical operators except for projection (including $\wedge$, $\vee$, $\neg$) are not parameterized so ENeSy could be trained with only link prediction task since meaningful complex query data might be hard to obtain in the real world.

The experimental results prove that our model outperforms existing complex query answering methods over standard knowledge graphs: FB15K-237 [18] and NELL-995 [22]. The performance, which is trained with link prediction, is also competitive to or better than the baselines trained with complex queries. The analysis of the model proves the effectiveness of neural and symbolic entanglement in solving the problem of cascading error and KG incompleteness with our framework.

## 2 Related Work

### 2.1 Knowledge Graph Embedding

A great deal of previous research into KG has focused on using machine learning to reason with the embedding method. The effects have been shown in TransE [5], TransH [20], TransG [21],

ComplEx [19], ConvE [8], DistMult [23] and RotatE [17]. These approaches aim to embed the entities and relations to a continuous vector space so that one-hop reasoning can be answered with link prediction. Different methods map the entities and relations into vector space with different distributions. Meanwhile, the rule and path-based models [10, 25, 24, 16], try to use learned patterns of path or rules to do multi-hop reasoning.

However, KGE models lack the ability to handle complex logical operators like conjunction ($\wedge$), disjunction ($\vee$), and negation ($\neg$), and they are not easy to generalize to multi-hop queries. In contrast, our framework can alleviate the problem of cascading error during multi-hop reasoning and also support all of the logical operators above so that any KGE models could be used to answer the complex query.

## 2.2 Embedding for Complex Query Answering

To answer logical queries, some works [9, 14, 26, 2, 15, 6] aims to encode the complex query into a space that proposes various geometric shape to represent the entity set and support complex query reasoning. Generally, a FOL query is converted to a computation graph with a directed acyclic graph (DAG) structure with which the query representation could be iteratively computed using the logical operation in embedding space. GQE (graph-query embedding) [9] consider conjunction operator ($\wedge$) with vector representation. Q2B (query to box) [14] replace the vector embedding with hyper-rectangles since it holds the idea that box embedding can represent sets of entities better and define the disjunctive norm form (DNF) to support the disjunction ($\vee$) operator. More recently, BetaE [15] models the query and entities with beta distributions which could support the negation ($\neg$) operator. CQD [1] applies beam search to an embedding model. ConE [26] proposes a new geometry model that embeds entities with cones embedding. FuzzQE [6] satisfies the axiomatic system of fuzzy logic to reason. Q2P (Query2Particles) [2] encodes each query into multiple vectors. GNN-QE [27] concentrates on the interpretation of the variables along the query path.

Most of these methods usually design their geometric or distribution embedding to support logical operators but the effectiveness of the projection operator, which is the most basic operator, has not been discussed. Meanwhile, they only concentrate on the final answers as labels but ignore the intermediate entities during the query process which are also important for reasoning.

## 3 Preliminaries

In this section, we introduce the task of complex query answering on KG. Given a set of entities $\mathcal{V}$ and a set of relations $\mathcal{R}$, a knowledge graph $\mathcal{G}$ is defined as $(\mathcal{V}, \mathcal{R}, \mathcal{T})$ where $\mathcal{T}$ is the set of triplets. A triplet is defined as $r(e_i, e_j)$ where $e_i \in \mathcal{V}, e_j \in \mathcal{V}, r \in \mathcal{R}$ if the relation $r$ exists between $e_i$ and $e_j$.

**First-Order Logic.** The purpose of complex query answering on KG is obtaining the target answer of FOL query which is defined with existential quantifiers($\exists$), conjunction ($\wedge$), disjunction ($\vee$), and negation ($\neg$). In a FOL query $q$, the anchor nodes are represented as a set $\mathcal{V}_a \subseteq \mathcal{V}$, existential quantified variables nodes are represented as $V_1, V_2, \ldots V_k$ and the target answer nodes is a variable $V_?$. Following the betaE [15], we use the FOL query in its disjunctive norm form, with which the query can be represented as a disjunction of several conjunctions. Finally, the query can be formulated as:

$$q[V_?] \coloneqq V_? : V_1, V_2, \ldots, V_k : c_1 \vee c_2 \vee \cdots \vee c_n.$$

where $c_i$ is a conjunction of several literals $a_{ij}$, i.e., $c_i = a_{ij} \wedge \cdots \wedge a_{im}$, and $a_{ij}$ is an atom or negation of an atom: $r(e_a, V)$ or $\neg r(e_a, V)$ or $r(V', V)$ or $\neg r(V', V)$ where $e_a \in \mathcal{V}_a$, $V, V' \in \{V_1, V_2, \ldots, V_k, V_?\}$ and $V \neq V'$ in an atom.

**Computation Graph and Logical Operators** As illustrated in figure 1, for a given FOL query, we can represent the whole process of reasoning as a computation graph. Each node corresponds to a variable $V$ or an anchor node $e_a$ and each edge represents a logical operation over the entity sets which includes the following operators:

- **Relational Projection:** Given an entity set $\mathcal{S} \subseteq \mathcal{V}$ and a relation $r \in \mathcal{R}$, the projection operator return a new entity set $\mathcal{S}'$ that contains the entities related to at least one of entity in $\mathcal{S}$: $\mathcal{S}' = \{e' \in \mathcal{V} | \exists r(e, e'), e \in \mathcal{S}\}$.

- **Intersection:** Given sets of entities $\{S_1, S_2, \ldots, S_n\}$ where $S_i \subseteq \mathcal{V}$, the intersection operator returns the intersection of these sets $\bigcap_{i=1}^n S_i$.

- **Union:** Given sets of entities $\{S_1, S_2, \ldots, S_n\}$ where $S_i \subseteq \mathcal{V}$, the union operator returns the union of these sets $\bigcup_{i=1}^n S_i$.

- **Complement:** Given a set of entities $\mathcal{S}$, the complement operator returns its complement $\mathcal{S}' = \mathcal{V} - \mathcal{S}$.

# 4 Methodology

In this section, we first introduce the neural and symbolic entangled projection operator in detail. Then we define symbolic-based logical operators and describe the ensemble prediction using embedding and symbolic results. Finally, the objective function and the learning procedure will be represented.

## 4.1 Neural and Symbolic Entangled Reasoning

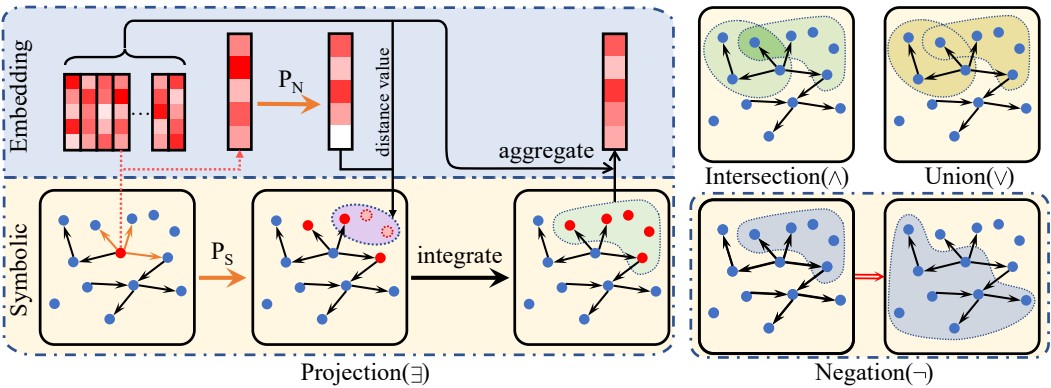

Figure 2: ENeSy's logical operators and the details about neural symbolic entanglement. $P_N$ means neural projection and $P_S$ means symbolic projection.

In our work, there are two ways to represent entities and relations. The neural part is just like the KGE or query embedding method we adopt, and rotatE [17] is chosen in this paper which embeds entities and relations to a complex space $\mathbb{C}^k$. For the symbolic part, an entity and a relation are encoded as a one-hot vector and an adjacent matrix, respectively.

Specifically, given the KG $\mathcal{G}$, entity set $\mathcal{V}$ and relation set $\mathcal{R}$, an entity $e$'s embedding representation is a vector $\mathbf{v}_e \in \mathbb{C}^k$ and its symbolic representation is encoded as a one-hot vector $\mathbf{p}_e \in \{0,1\}^{1 \times |\mathcal{V}|}$. Each relation $r$ is modeled as a vector $\mathbf{v}_r \in \mathbb{C}^k$ and the corresponding symbolic representation is an adjacent matrix $\mathbf{M}_r \in \{0,1\}^{|\mathcal{V}| \times |\mathcal{V}|}$ where $\mathbf{M}_r^{ij} = 1$ if $r(e_i, e_j) \in \mathcal{G}$, else $\mathbf{M}_r^{ij} = 0$.

**Neural projection** follows the embedding method, we use rotatE [17] here as an example. For a projection query $(h, r, ?)$, the functional mapping with relation $r$ is an element-wise rotation from $h$ to the target answer $t$:

$$\mathbf{v}_t = \mathbf{v}_h \circ \mathbf{v}_r, \text{where} |\mathbf{v}_r| = 1 \tag{1}$$

and $\circ$ means Hadamard product. This step gets the predicted embedding which could be used to search for the target answers.

**Symbolic projection** is conducted with matrix multiplications followed `TensorLog` [7]:

$$\mathbf{p}_t = \mathbf{g}(\mathbf{p}_h \mathbf{M}_r)^\top \tag{2}$$

where $\mathbf{p}_t$ is a multi-hot vector that represents the entities linked with $h$ via relation $r$, $\top$ means transposing the vector and $\mathbf{g}$ is a normalization function. Particularly, we consider the function as $\mathbf{g}(\mathbf{x}) = \mathbf{x}/sum(\mathbf{x})$. This step gets the target entities by traversing KG.

**Entangled projection** tries to take advantage of these above two results. Since embedding prediction suffers from the cascading error but could obtain the answer not linked with $h$, while traversing KG could not get the answers which lack the edges to the head entity but the searched answers are convincing, ENeSy combines them in an entanglement way as illustrated in Figure 2. Specifically, the similarity between the predicted embedding $\mathbf{v}_t$ and all the entities are calculated first. We define the similarity function as:

$$\mathbf{S}(\mathbf{x}, \mathbf{y}) = \gamma - \|\mathbf{x} - \mathbf{y}\|_1 \tag{3}$$

where $\gamma$ denotes the margin. The L1 norm function can be changed to any distance function. We define a new vector representation with these similarity values. After softmax function, we get a inferred vector $\mathbf{p}'_t \in [0, 1]^{1 \times |\mathcal{V}|}$ which is generated from $\mathbf{v}_t$. With $\mathbf{p}'_t$, a new symbolic vector $\mathbf{p}''_t$ is obtained by the following step:

$$\mathbf{p}''_t = \mathbf{g}(\mathbf{p}_t + \mathbf{p}'_t) \tag{4}$$

This new symbolic vector $\mathbf{p}''_t$ integrates the information from the embedding $\mathbf{v}_t$ with symbolic reasoning results $\mathbf{p}_t$. This procedure adds more entities, which might be the answer to the query which are not linked with $h$, to $\mathbf{p}''_t$. This enhancement eliminates the limitation of KG incompleteness of symbolic reasoning to some extent. Each element of $\mathbf{p}''_t$ could be regarded as the probability of the corresponding entity. Let's assume the entity set with non-zero probability as $\mathcal{S}_t$, a aggregation function is employed to transfer the symbolic vector $\mathbf{p}''_t$ and the embedding of entities in $\mathcal{S}_t$ to a new embedding vector $\mathbf{v}'_t$ with an MLP function:

$$\mathbf{v}'_t = \sum_{i=1}^{|\mathcal{S}_t|} \mathbf{p}_t^{i''} \mathrm{MLP}(\mathbf{v}_{e_i}) \mathbf{v}_{e_i}, e_i \in \mathcal{S}_t \tag{5}$$

where $\mathbf{p}_t^{i''}$ is the corresponding probability of $e_i$ in $\mathbf{p}_t$. This new embedding $\mathbf{v}'_t$ aggregates the symbolic answer, Note that although we use the rotatE as the neural function, any other sufficiently expressive KG embedding model or the projection operator of any other query embedding models could be employed with our framework in theory.

## 4.2 Neural and Symbolic Ensemble Answering

With the symbolic vector $\mathbf{p}$, the logical operator intersection($\mathbf{p}_1 \wedge \mathbf{p}_2$), union($\mathbf{p}_1 \vee \mathbf{p}_2$) and negation ($\neg\mathbf{p}$) could be defined as follows:

$$\mathbf{p}_1 \wedge \mathbf{p}_2 : \mathbf{g}(\mathbf{p}_1 \circ \mathbf{p}_2), \quad \mathbf{p}_1 \vee \mathbf{p}_2 : \mathbf{g}(\mathbf{p}_1 + \mathbf{p}_2 - \mathbf{p}_1 \circ \mathbf{p}_2), \quad \neg\mathbf{p} : \mathbf{g}(\frac{\alpha}{|\mathcal{V}|} - \mathbf{p})$$

where $\circ$ is the Hadmard product and $\alpha$ is a hyperparameter. After getting symbolic vector with these logical operators, the MLP function used in Equation (5) is employed to get an aggregated embedding.

The embedding and symbolic vector can both be used to get the final answers. For the neural part, the similarity between the embedding vector $\mathbf{v}$ and all the entities $e \in \mathcal{V}$ is computed with Equation (3), which is used to rank the candidate answers. For the symbolic part, since the vector $\mathbf{p}$ represents the probability of each entity, the answers can be directly obtained with ranked elements of $\mathbf{p}$. To make the ensemble using the two type answers, we set $\lambda$ to get a combined result with $\mathbf{v}$ and $\mathbf{p}$ as:

$$\mathbf{a} = \lambda\mathbf{p} + (1 - \lambda)\mathrm{Softmax}(\underset{\forall e \in \mathcal{V}}{\mathbf{Concat}}(\mathbf{S}(\mathbf{v}, \mathbf{v}_e))) \tag{6}$$

where $\lambda$ is the weight to balance the influence of $\mathbf{v}$ and $\mathbf{p}$ and $\mathbf{Concat}$ is a function mapping the similarity between all entities $e \in \mathcal{V}$ and $\mathbf{v}$ to a vector. The final answers can be determined with $\mathbf{g}(\mathbf{a}) \in [0, 1]^{1 \times |\mathcal{V}|}$.

## 4.3 Learning Procedure

Given a query $q$ with answer entity set $\mathcal{S}_q$, after getting the final answer embedding $\mathbf{v}_q$ and symbolic vector $\mathbf{p}_q$, we construct two following objective loss functions:

$$L_1 = -\log\sigma(-\mathbf{S}(\mathbf{v}_q, \mathbf{v}_e)) - \frac{1}{n}\sum_{i=1}^{n} \log\sigma(\mathbf{S}(\mathbf{v}_q, \mathbf{v}_{e'})) \tag{7}$$

$$L_2 = -\log\sigma(\mathbf{p}_e \cdot \log[\mathbf{p}_q^\top, \theta]_+) \tag{8}$$

where $e \in \mathcal{S}_q$ is an answer of $q$, $e' \notin \mathcal{S}_q$ is a negative answer which is sampled randomly. $\sigma$ is the `sigmoid` function, $\cdot$ is dot-product and $[\mathbf{x}, \theta]_+$ denotes the maximum value between each element of $\mathbf{x}$ and $\theta$, which is a threshold.

Meanwhile, the MLP function in Equation (5) needs to be pretrained individually. Since this function is employed to convert a symbolic vector to an embedding vector, and in the projection operator, we have $\mathbf{p}'_t$ which is generated from $\mathbf{v}_t$, MLP could be used to convert $\mathbf{p}'_t$ back to $\mathbf{v}_t$. Based on this, we also design a loss function to train this function as follows:

$$L_3 = -\log\sigma(-\mathbf{S}(\mathbf{v}_t, \sum_{i=1}^{|\mathcal{S}'_t|} \mathbf{p}_t^{i\prime}\text{MLP}(\mathbf{v}_{e_i})\mathbf{v}_{e_i})), e_i \in \mathcal{S}'_t \tag{9}$$

where $\mathcal{S}'_t$ is the corresponding entity set whose probabilities are not zero in $\mathbf{p}'_t$. In the first step of the training process, the symbolic part is not included, and only the embedding of entities and relations will be trained with the link prediction task. Next, in the second step, the completed projection operator will be trained still by link prediction and the loss function is $L = L_1 + L_2 + L_3$. In theory, the model could answer complex queries after the above two steps, but it could also be fine-tuned with complex query data using the loss function $L = L_1 + L_2$.

## 5 Experiment

In this section, we evaluate the ability of ENeSy on answering the complex query on several KG benchmark datasets. The experiment results demonstrate that: 1) The performance of ENeSy is excellent; 2) We can train ENeSy with only link prediction task to answering complex queries; 3) The embedding and symbolic parts of our model can enhance each other.

### 5.1 Dataset and Experiment Setting

#### 5.1.1 Datasets and Evaluation Protocol

We perform the experiments on two benchmarks, FB15K-237 [18] and NELL-995 [22]. FB15K-237 is a subset from Freebase [3] and removes the inverse relation. NELL-995 is a dataset constructed from high-confidence facts of NELL [12].

The focus of our experiment is answering FOL queries on incomplete KGs so we only evaluate the ability of models to obtain the answers that could not be discovered by traversing KGs. Specifically, with the standard training, validation, and testing set, the edges in KG could be divided into three parts, which are training edges, validation edges, and test edges. The corresponding graph $\mathcal{G}_{train}$, $\mathcal{G}_{valid}$ and $\mathcal{G}_{test}$ are build with *training* edges, *training + validation* edges and *training + validation + test* edges, respectively, so we have $\mathcal{G}_{train} \subset \mathcal{G}_{valid} \subset \mathcal{G}_{test}$. We only use the queries whose answer sets $\mathcal{A}_{train}$, $\mathcal{A}_{valid}$ and $\mathcal{A}_{test}$ on different graph have $\mathcal{A}_{train} \subset \mathcal{A}_{valid} \subset \mathcal{A}_{test}$. The answers in $\mathcal{A}_{valid} - \mathcal{A}_{train}$ could be used to tune the hyper-parameters and results are reported with the answer entities in $\mathcal{A}_{test} - \mathcal{A}_{valid}$. This means we only evaluate on entities that are not the answers to the training query set and the model has not seen them. Meanwhile, these answers could not be found with graph traversal on $\mathcal{G}_{train}/\mathcal{G}_{valid}$. For each answer of a test query, we calculate the Mean Reciprocal Rank (MRR) as evaluation metrics, and the results are reported with filter setting as TransE [5], in which all other correct answers are filtered out before calculating the rankings of answers.

The queries sampled from these two benchmarks are provided by BetaE [15] which is an expansion of the version provided by Q2B [14]. The query set contains 14 different types of query structures are shown in Figure 3. In order to further verify the generalization ability of the models, only 5 conjunctive queries (*1p/2p/3p/2i/3i*) and 5 query types with negation (*2in/3in/inp/pni/pin*) are used to train the model, while the other four query types (*2u/up/ip/pi*) do not appear in the training process and are directly evaluated when testing, which makes this task more challenging.

#### 5.1.2 Baseline

We consider four baselines as the compared methods in the following sections: Graph Query Embedding (GQE) [9] embeds the query into vectors, which could handle the projection and conjunction

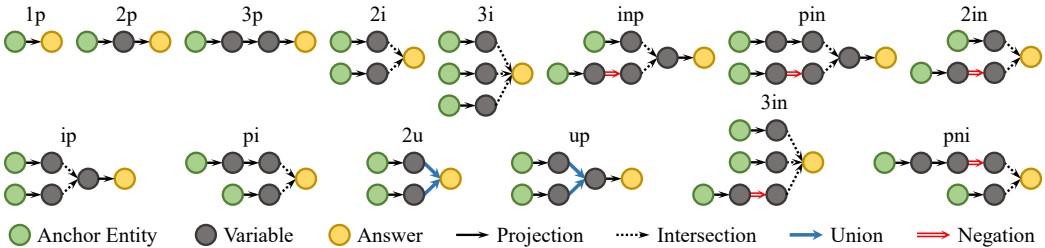

Figure 3: The query structure of all queries used for training and evaluation. Namely, the *p*, *i*, *u* and *n* stands for the projection, intersection, union and negation, respectively.

Table 1: The MRR results of FOL queries on FB15K-237 and NELL-995, and the models are trained with only link prediction task. The $\text{Avg}_p$ and $\text{Avg}_n$ are the average MRR of Existential Positive First Order (EPFO) queries (query with $\exists$, $\vee$ or $\wedge$ and without $\neg$) and queries with $\neg$, respectively. N/A means not available.

| Model | $\text{Avg}_p$ | $\text{Avg}_n$ | 1p | 2p | 3p | 2i | 3i | pi | ip | 2u | up | 2in | 3in | inp | pin | pni |
|---|---|---|---|---|---|---|---|---|---|---|---|---|---|---|---|---|
| | | | | | | FB15K-237 | | | | | | | | | | |
| GQE | 17.7 | N/A | 41.6 | 7.9 | 5.4 | 25.0 | 33.6 | 16.3 | 10.9 | 11.9 | 6.2 | N/A | N/A | N/A | N/A | N/A |
| Q2B | 18.2 | N/A | 42.6 | 6.9 | 4.7 | 27.3 | 36.8 | 17.5 | 11.1 | 11.7 | 5.5 | N/A | N/A | N/A | N/A | N/A |
| BetaE | 15.8 | 0.5 | 37.7 | 5.6 | 4.4 | 23.3 | 34.5 | 15.1 | 7.8 | 9.5 | 4.5 | 0.1 | 1.1 | 0.8 | 0.1 | 0.2 |
| FuzzQE | 21.8 | 6.6 | 44.0 | **10.8** | **8.6** | 32.3 | 41.4 | 22.7 | 15.1 | **13.5** | **8.7** | 7.7 | 9.5 | 7.0 | 4.1 | 4.7 |
| **ENeSy** | **23.4** | **8.1** | **44.5** | **10.8** | 7.7 | **33.2** | **48.4** | **25.8** | **18.8** | 13.4 | 7.6 | **9.6** | **10.2** | **7.1** | **5.8** | **7.8** |
| | | | | | | NELL-995 | | | | | | | | | | |
| GQE | 21.7 | N/A | 47.2 | 12.7 | 9.3 | 30.6 | 37.0 | 20.6 | 16.1 | 12.6 | 9.6 | N/A | N/A | N/A | N/A | N/A |
| Q2B | 21.6 | N/A | 47.6 | 12.5 | 8.7 | 30.7 | 36.5 | 20.5 | 16.0 | 12.7 | 9.6 | N/A | N/A | N/A | N/A | N/A |
| BetaE | 19.0 | 0.4 | 53.1 | 6.0 | 3.9 | 32.0 | 37.7 | 15.8 | 8.5 | 10.1 | 3.5 | 0.1 | 1.4 | 0.1 | 0.1 | 0.1 |
| FuzzQE | 27.1 | 7.3 | 57.6 | 17.2 | **13.3** | 38.2 | 41.5 | 27.0 | 19.4 | **16.9** | **12.7** | 9.1 | **8.3** | 8.9 | 4.4 | 5.6 |
| **ENeSy** | **28.7** | **9.4** | **58.8** | **17.4** | 12.8 | **39.1** | **48.9** | **29.1** | **24.1** | 16.0 | 12.4 | **10.9** | 8.2 | **11.0** | **8.4** | **8.6** |

queries. With DNF setting, it could also support the union operator. Query2Box (Q2B) [14] uses box embedding to represent the query and entity sets and could answer existential positive first-order (EPFO) logic queries. Beta Embedding (BetaE) [15] models query as Beta Distributions which enables it to support negation($\neg$) operation. CQD [1] applies beam search to an embedding model but it could not support the negation operator. FuzzQE [6] uses fuzzy logic to embed the query.

The MRR results of these baselines are from the BetaE [15] and FuzzQE paper [6]. The first two methods can't handle negation operation and among these models, only FuzzQE can be trained with only link prediction tasks and answer the complex query.

### 5.1.3 Training Procedure and Experiment Settings

To train the model, we first only train the embedding of relations and entities with *1p* queries which is similar to the pure KG embedding training. Second, the projection operator of ENeSy is trained on 1p queries. Meanwhile, the queries of all structures can be used to fine-tune the model.

We implement our model with Pytorch framework and train our model on RTX3090 GPU. The ADAM optimizer was used to parameter tune with a learning rate of 0.0001 that will decrease during the training process. In the second step, the learning rate is set to be $10^{-5}$ to train the model with $1p$ queries, and in the fine-tuning process, the learning rate starts with $2 * 10^{-7}$. We set the embedding dimension of the entity and relation to 1024, respectively. The hidden state dimension of MLP is 1024. The training batch size is $\{64, 16\}$ for FB15K-237 and NELL-995, while the negative sample size is $\{128, 32\}$. The margin $\gamma$ used in similarity computation is 24. $\theta$ used as a threshold is $10^{-10}$. $\alpha$ is set to be 10. The choices of $\lambda$ which is used for ensemble prediction are based on the results of *valid* set for each query type. The best hyperparameter setting is selected by the MRR metric on the *valid* set. We run the experiment several times and find random seed has almost no effect on the result, but in the fine-tuning process, the learning rate influence the accuracy.

Table 2: The average MRR results of FOL queries on FB15K-237 and NELL-995 , and the models are trained with complex query data.

| Model | Avg$_p$ | Avg$_n$ | 1p | 2p | 3p | 2i | 3i | pi | ip | 2u | up | 2in | 3in | inp | pin | pni |
|---|---|---|---|---|---|---|---|---|---|---|---|---|---|---|---|---|
| | | | | | | | FB15K-237 | | | | | | | | | |
| GQE | 16.3 | N/A | 35.0 | 7.2 | 5.3 | 23.3 | 34.6 | 16.5 | 10.7 | 8.2 | 5.7 | N/A | N/A | N/A | N/A | N/A |
| Q2B | 20.1 | N/A | 40.6 | 9.4 | 6.8 | 29.5 | 42.3 | 21.2 | 12.6 | 11.3 | 7.6 | N/A | N/A | N/A | N/A | N/A |
| BetaE | 20.9 | 5.5 | 39.0 | 10.9 | 10.0 | 28.8 | 42.5 | 22.4 | 12.6 | 12.4 | 9.7 | 5.1 | 7.9 | 7.4 | 3.5 | 3.4 |
| CQD | 21.7 | N/A | **46.3** | 9.9 | 5.9 | 31.7 | 41.3 | 21.8 | 15.8 | 14.2 | 8.6 | N/A | N/A | N/A | N/A | N/A |
| FuzzQE | 24.2 | **8.5** | 42.2 | **13.3** | **10.2** | 33.0 | 47.3 | 26.2 | 18.9 | **15.6** | **10.8** | 9.7 | **12.6** | **7.8** | 5.8 | 6.6 |
| **ENeSy** | **24.5** | **8.5** | 44.7 | 11.7 | 8.6 | **34.8** | **50.4** | **27.6** | **19.7** | 14.2 | 8.4 | **10.1** | 10.4 | 7.6 | **6.1** | **8.1** |
| | | | | | | | NELL-995 | | | | | | | | | |
| GQE | 18.6 | N/A | 32.8 | 11.9 | 9.6 | 27.5 | 35.2 | 18.4 | 14.4 | 8.5 | 8.8 | N/A | N/A | N/A | N/A | N/A |
| Q2B | 22.9 | N/A | 42.2 | 14.0 | 11.2 | 33.3 | 44.5 | 22.4 | 16.8 | 11.3 | 10.3 | N/A | N/A | N/A | N/A | N/A |
| BetaE | 24.6 | 5.9 | 53.0 | 13.0 | 11.4 | 37.6 | 47.5 | 24.1 | 14.3 | 12.2 | 8.5 | 5.1 | 7.8 | 10.0 | 3.1 | 3.5 |
| CQD | 28.4 | N/A | **60.0** | 16.5 | 10.4 | **40.4** | 49.6 | 28.6 | 20.8 | 16.8 | 12.6 | N/A | N/A | N/A | N/A | N/A |
| FuzzQE | 29.3 | 8.0 | 58.1 | **19.3** | **15.7** | 39.8 | **50.3** | 28.1 | 21.8 | **17.3** | **13.7** | 8.3 | **10.2** | 11.5 | 4.6 | 5.4 |
| **ENeSy** | **29.4** | **9.8** | 59.0 | 18.0 | 14.0 | 39.6 | 49.8 | **29.8** | **24.8** | 16.4 | 13.1 | **11.3** | 8.5 | **11.6** | **8.6** | **8.8** |

## 5.2  Trained with Link Prediction

Since meaningful FOL queries are usually not available in real scenes, we first report the results of models which are trained with only *1p* query data, which also can be seen as a link prediction task. The MRR results are shown in Table 1.

As the table shows, compared with pure embedding methods like GQE, Q2B, and BetaE, the performance of ENeSy significantly improves. Since the operators except for the projection of these methods are parameterized, the ability to handle complex queries is limited, but our model can be generalized to more complex query structures. Even though, the improvement on *1p* query proves the advantages of generalizing other KG embedding models to solve this problem. Compared with FuzzQE, which could also be trained with link prediction, our model improves the average MRR of EPFO by about 2.6%(relatively 11.9%) on FB15K-237 and 1.6%(relatively 5.9%) on NELL-995. For the queries with negation, ENeSy provides a more absolute improvement, which is 1.5%(relatively 24.6%) and 2.1%(relatively 28.8%) on FB15K-237 and NELL-995. We believe the reason for this enhancement is that the symbolic representation can indicate the probability of each entity better.

## 5.3  Trained with Complex Query

The results of models trained with query data are also reported in Table 2. Compared with the average MRR in Table 1, the performance of most baselines improves a lot because of the labeled data of complex queries, but ENeSy still achieves the best performance on the average MRR of EPFO query and negation query. Moreover, our model could get closer results with different training settings. Specifically, the average MRR decreases by about 4.7% and 2.4% for EPFO and 5% and 4.1% for negation queries on FB15K-237 and NELL-995 respectively when trained with link prediction. For FuzzQE, the results decline about 9.9%, 7.5% for EPFO, and 22.4%, 8.8% for query with negation on FB15K-237 and NELL-995, respectively. This proves the stronger robustness of our methods. Note that though the *2p/3p* results seem to be worse than FuzzQE, we can replace RotatE with the projection operator of FuzzQE, and the results should be the same in theory. All the models do not get good results on the negation query. We think it's because, after the negation operation, most of the entities in KG will be included, which makes the reasoning hard. Maybe a better way is only considering the entities which belong to the same type as the entities before the negation operation.

## 5.4  Analysis of ENeSy

To get deep insights into the neural and symbolic reasoning part of ENeSy, we investigate their impact in this section. In what follows, we first explore how symbolic affects the embedding results. We then examine the influence of embedding on symbolic. Finally, the effect of ensemble prediction is discussed. The results are based on FB15K-237, and the situation on NELL-995 is similar.

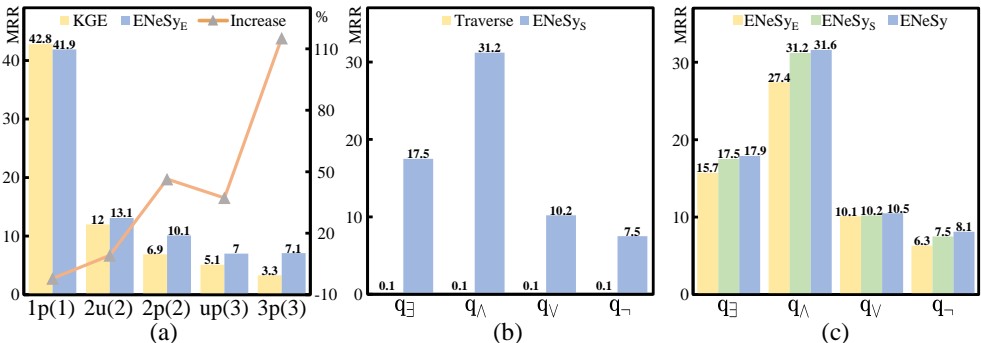

Figure 4: The figure of model analysis. (a): The MRR results and increase scale of ENeSy$_E$ and pure KGE models. The query types are sorted by the query length. (b): The average MRR results of ENeSy$_S$ and traversing. The queries are grouped by their operator. q$_\exists$ includes all of the query type, q$_\wedge$ includes *2i/3i/pi/ip*, q$_\vee$ includes *2u/up* and q$_\neg$ includes *2in/3in/inp/pin/pni*. (c): The average MRR results of ENeSy$_E$, ENeSy$_S$ and ENeSy, queries are grouped in the same way as (b).

### 5.4.1 Q1: Do symbolic results assist neural reasoning in cascading error?

We compare the pure KE embedding model, which is RotatE in this experiment, with the embedding results of ENeSy without ensemble using, which we denote as ENeSy$_E$. For fairness of training data, after the KGE model has converged which is trained with *1p* query, ENeSy$_E$ is trained with *1p* data rather than complex query data based on the KGE model. Since the KGE method could not support logical operators, the results of *1p/2p/3p* query and the *2u/up* with DNF are reported. The MRR results and the ratio of improvement are shown in Figure 4 (a). The query types are listed below the horizontal axis and we sort them by the length of the query which is the longest distance from the anchor nodes to the target node in the computation graph, and it's marked after the query type. Based on the similar performance on *1p* query, the MRR results of more complex queries significantly improve with query length increases. This comparison demonstrates that the cascading error, which is the main limitation of multi-hop embedding reasoning, has been alleviated with the symbolic assistant.

### 5.4.2 Q2: Does embedding results assist symbolic reasoning in KG incompleteness?

The symbolic MRR results of ENeSy without ensemble, denoted as ENeSy$_S$, and pure symbolic results are shown in Figure 4 (b). We divided the query types into four groups according to the operator they have. Since we only evaluate the generalization ability of models with answers that could not be found by simply traversing KG, the traversing results are nearly zero (since the result is MRR, the number won't be an absolute zero), while the ENeSy$_S$ achieve better results than most baselines. The reason for this significant improvement from zero to almost SOTA performance is in the entangled process, ENeSy successfully captures the information from embedding which makes symbolic results include the answers that could not be obtained directly. In summary, the experiment proves that the KG incompleteness problem for symbolic reasoning can be solved in our framework.

### 5.4.3 Q3: Is ensemble prediction of neural and symbolic results useful?

Ensemble prediction enables us to fuse the symbolic and reasoning results. To verify its effectiveness, we compare the performance of ENeSy$_E$, ENeSy$_S$ and ENeSy trained with *1p* queries. The MRR results are shown in Figure 4 (c). The queries grouped in the same way as Q2. As the figure illustrates, all the results of different group queries improve with ensemble using. Specifically, the average MRR of the different groups increased by about 14.0%, 15.3%, 4.0%, 28.6% and 2.3%, 1.3%, 2.9%, 8.0% compared with ENeSy$_E$ and ENeSy$_S$, respectively, which certifies that the two parts could not only enhance each other in the reasoning process but also can be combined in the final results.

## 6 Conclusion

In this paper, we proposed ENeSy, a neural-symbolic entangled framework for answering complex logical queries over KGs. This model could generalize any embedding methods to the complex query and use the symbolic reasoning results to alleviate cascading error, while the symbolic part also benefits from neural reasoning to solve the problem of KG incompleteness. The ENeSy supports all the FOL operations and can be trained with only link prediction tasks. Experimental results show that our model achieves state-of-the-art performances in answering FOL queries with strong robustness, and each part is tightly entangled to enhance the other.

## Acknowledgements

This work is funded by NSFCU19B2027/91846204.

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
