# OpenReview forum: "Neural-Symbolic Entangled Framework for Complex Query Answering"
_NeurIPS.cc/2022/Conference — NeurIPS 2022 Accept_

### Official Review · Reviewer_7Bea · 2022-06-30

**Rating:** 7
**Confidence:** 3
**Soundness:** 3 good
**Presentation:** 4 excellent
**Contribution:** 3 good

**Summary:**

This paper proposes a neural-symbolic approach (called ENeSy) for answering FOL-based queries over KGs. The main promise of the proposed approach lies in its ability to alleviate the problem of cascading error as it existed in previous approaches and also the incompleteness of KGs.

As claimed in the paper, existing query embedding methods handle these complex queries over incomplete KB but they don’t pay any attention to intermediate entities which are important for answering multi-hop queries. The proposed approach ENeSy seems to bridge some of these gaps.

Overall, ENeSy comprises two parts (neural and symbolic)  to represent a given KG.
- For the neural part, it uses rotatE to embed entities and relations into a complex Space ${\mathbb{C}}^k$.
- For the symbolic part, an entity and a relation are encoded as a one-hot vector and an adjacent matrix, respectively.
- The key operations involved in answering complex queries are _projection, intersection, union,_ and _negation_.  ENeSy proposes an entangled version of each of these operations in Equations (5) and (6).

The experiments are performed on two datasets (FB15K-237 and NELL-995). 	The focus of the experiments is on answering FOL queries on incomplete KGs so as to evaluate the ability of models to obtain the answers that could not be discovered by traversing KGs. The experiment results (Table 1 and 2) and the insights (Figure 4) established the merits of the proposed ENeSy over the baselines.

**Questions:**

- In lines 162 and 168, why the word _possibility_ is used and not the word _probability_?
- How does MLP(.) function look like in equation (5)? Does it map a vector to a scalar? If it is a vector then I am not sure how this vector is getting multiplied with the other two vectors present in equation (5)? Is it an inner product between two vectors? Notations are a bit confusing as there is no transpose sign.
- In equation (7), should it be $v_q$ instead of $v$?
- For baselines, did you recompute the numbers or picked from the existing papers? If picked from the existing papers, whether the existing papers followed the same evaluation steps as mentioned in lines # 205-217? Because, otherwise, it will not be an apple-to-apple comparison.


**Limitations:**

There is not much of a discussion on the limitations of the proposed approach. I would have loved to see some discussion on the memory footprints and time complexity-related issues and how they compare with baselines. Also, it may be worth discussing if there is any class of queries within FOL that are not being handled properly by your approach? By looking at Figure 4(b), it looks like negation queries are relatively hard to deal with. You can have some discussion about that as well.

**Strengths And Weaknesses:**

**Originality**

The idea of fusing neural and symbolic representations of the answer entities to cope with the KG incompletion is certainly novel. Neural and symbolic representations seem to be helping each other and thereby boosting the overall performance.


**Quality**

The paper is really very well written. The entanglement ideas are novel. The experimentation evaluation plan laid out in lines # 205-217 is very clearly written and makes perfect sense.  The experiments seem to be convincing in terms of proving the merits of the proposed approach. The only reservation I have is that the experiments are performed only on two datasets. Authors could have included one more dataset.


**Clarity**

By and large paper is easy to read and follow through. Below are some questions regarding the clarity of the matter.
- In line # 116, the symbol S’ seems a bit inconsistent.
- In line # 139, the spelling of Hadamard is incorrect.
- In lines 162 and 168, why the word _possibility_ is used and not the word _probability_?
- How does MLP(.) function look like in equation (5)? Does it map a vector to a scalar? If it is a vector then I am not sure how this vector is getting multiplied with the other two vectors present in equation (5)? Is it an inner product between two vectors? Notations are a bit confusing as there is no transpose sign.
- In equation (7), should it be $v_q$ instead of $v$?
- Caption of Table 1 seems to have typos. Please read it carefully once again and fix any typos.
- Just to make the paper self-contained, better explain the convention used behind assigning titles to the computation graphs in Figure 3. For example, in the case of 2i, why can’t we call it 2p2i. or why “inp” can’t be titled as “pin”. If need be, add this in the appendix.
- In Tables 1 and 2, you should also tag the count of each query type in your test set over which these numbers are reported.

**Significance**

If we were to believe the numbers shown in Tables 1 and 2 then it clearly proves a significant improvement over the baselines methods. However, as I mentioned before, it would have been nicer to run experiments on one more dataset. Nonetheless, what is presented is still impressive.

---

> ### Author Response · Authors · 2022-08-02
> **Response to Reviewer 7Bea**
>
> Thank you for your very careful review of our paper, and for the comments, corrections and suggestions that ensued. We fix the writing typos according to your opinions. Also, we update a new version of this paper, some statistics you suggested are added in the Appendix.
>
> For  Quality:
> - We regard FuzzQE, which could be trained with only 1p query, as the main baseline. Thus we follow the experiments in FuzzQE reporting results on FB15K-237 and NELL-995 and training with all queries and only 1p query. For the completeness of the table and the limitation of paper space, we didn’t report the results on FB15K.
>
> For Clarity:
> - Clarity 1, 2, 6: We will fix them and thanks for your careful reading again.
> - Clarity 3: _probability_ is a better choice and we will change the word.
> - Clarity 4: The MLP function is used to map a vector to a scalar. The $\mathbf{p}_{t}^{i\prime\prime}$ in equation (5) is the corresponding probability of an entity, and it’s a scalar, too. The equation means a vector is multiplied with two scales.
> - Clarity 5: It should be $\mathbf{v}_{q}$ rather than $\mathbf{v}$.
> - Clarity 7: We just employ the names used in previous works.
> - Clarity 8: We add the numbers of different queries in the Appendix.
>
> For Question:
> - Question 1, 2, 3 refers to replies to Clarity 3, 4, 5.
> - For Question 4: The baseline results are from the existing papers with the same evaluation protocol.
>
> For limitation:
> - We add an experiment about the time(millisecond) of answering the test queries:
> |Operation|ENeSy|BetaE|
> |:-------:|:---:|:---:|
> |Time_p|13.5|5.8|
> |Time_n|16.3|5.9|
>
>   _Time_p_ and _Time_n_ are the time of average time of answering queries with and without negation operator in the test set, respectively. Our model takes more time than pure query embedding methods since we use two different methods to reason, and we think the growth in time is foreseeable and acceptable.
> - For the concern about memory footprints, though it can be trained with a RTX3090(24GB), ENeSy needs more space than baselines because of  the huge adjacent matrices of relations with dimension of __(#relation, #entity, #entity)__, but over 99% of the elements are __0__. In theory, this could be solved by using a sparse way to represent the matrices. However, Pytorch doesn’t support sparse matrices well for now, and we believe this won’t be a limitation in the near future.
> - For the concern about the performance of negation queries, all the models handling FOL query can't do well. We think it’s because after the negation operation, most of the entities in KG will be regarded as answers, which makes the following reasoning hard. Maybe a better way is only considering the entities which belong to the same type with the entities before negation operation. We add these discussions in the new version.

---

> > ### Comment · Reviewer_7Bea · 2022-08-10
> > **Thank you.**
> >
> > Thank you for your response. I don't have further questions at the moment.

---

### Official Review · Reviewer_3t6R · 2022-07-10

**Rating:** 5
**Confidence:** 4
**Soundness:** 3 good
**Presentation:** 3 good
**Contribution:** 3 good

**Summary:**

In this paper, the authors present a new method to answer complex logical queries on Knowledge Graphs. They include symbolic logic to answer FOL queries and take into account intermediate answers to the queries. By adding the neural symbolic model, the paper aims to tackle some of the main disadvantages of the best-known embedding methods: cascading error and KG incompleteness.

**Questions:**

1. Have you tested the entangling framework with other embedding methods? Why did you select rotatE?
2. As far as I understand, the entangling method only adds more possible answers to a query, have you checked in which percentage this number increases?
3. Have you manually checked some of the queries you are using for training and testing purposes?


**Limitations:**


The main advantage of this new entangling framework is that it can be added to any other existing framework. However, how much improvement it actually provides, in general, is still unclear.


**Strengths And Weaknesses:**

Strengths
- The authors clearly define their methodology and experimentation.
- The paper combines previous technologies in a novel way that achieves better results. The experiments prove the proposed method achieves better results as the queries get more complex.
- The proposed entangling framework can include any other embedding method. Thus, in theory, it adds a performance improvement to all previous KGE methods.

Weaknesses
- My main concern is the fact that the entangling framework adds more possible answers to the queries. I would like to see a comparison of how many of the possible answers added by the symbolic logic are actually correct, or else it is highly increasing the number of False Positives.
- According to the results, everything points in the direction that any other embedding method can have similar performance improvements. But I am missing some experimentation in that direction to show how each well-known method is improved by adding this entangling framework.
- Important baselines are missing. On many of the metrics, CQD[a] can achieve better results than that of the proposed model. The recent model GNN-QE[b] outperforms the proposed model almost on all metrics. These two models are closely related to this work but not discussed.
- Performances on FB15k are missing.

[a] Complex query answering with neural link predictors, ICLR 2021
[b] Neural-Symbolic Models for Logical Queries on Knowledge Graphs, ICML 2022

---

> ### Author Response · Authors · 2022-08-02
> **Response to Reviewer 3t6R**
>
> Thanks very much for the constructive comments. In the following we group related weaknesses and questions and address them point by point. We hope our replies can answer your questions. Also, we update a new version of the paper and add some experiments you suggested.
>
> For Weakness:
> - For Weakness1 & Question2: The entangling framework doesn’t simply ‘add more possible answers to the queries’ since the evaluation metric (i.e., MRR) is in a ranking setting over all entities where the number of candidate answers is fixed. The entangling framework helps score all candidate entities in better order, making correct entities ranked higher.\
> Moreover, we count the entities with probability larger than one-tenth of the largest element in $\mathbf{p}_t^{\prime\prime}$ (the vector after entanglement) and $\mathbf{p}_t$ (the vector before entanglement) for 1p/2p/3p on the last step, and these entities with higher probability can be seen as ‘right answers’ in this experiment. Compared to the numbers before entanglement, they increase __6%, -20% and -27%__, respectively. The numbers decrease on 2p/3p since we set a threshold to filter the entities, and this result means the model doesn’t simply 'add more answers', but also enables a limited number of more accurate answers to get more attention.
> - For Weakness2 & Question1: Yes, any other embedding method with a projection function could be applied in our framework and is expected to have similar improvements in theory.
> Due to space limitation, we choose one embedding method, RotatE, to demonstrate the effectiveness of our framework, since RotatE is proved to be simple and effective at projecting one entity to other entities in vector space given a relation. To verify the generalization ability of our framework, we also test another typical embedding method, TransE.  Specifically, we train TransE with 1p queries (link prediction). The query types that are supported by embedding models are sorted by query length. Based on the embeddings, we add the entanglement framework, and note that we don’t train it with complex query data for fairness. Results on FB15K-237 are as follows, which shows that as the query gets longer, the MRR results improve more. The tendency is similar to RotatE (Refer to Figure 4(a)).
> ||1p|2u|2p|up|3p|
> |-|-|-|-|-|-|
> |TransE|39.33|9.79|7.04|5.49|5.08|
> |ENeSy(TransE)|41.03|12.06|11.03|8.58|9.38|
> |Improvement(%)|4.3|23.2|56.7|56.3|84.6|
> - For Weakness3: First, CQD[2] can not support the __negation__ operator, and we compare our methods with general models (BetaE, FuzzQE) that can handle the negation operator, or typical models(Q2B, GQE).  Meanwhile, CQD is the main baseline in FuzzQE[3], which has proved that FuzzQE achieves better performance than CQD, and FuzzQE is our main baseline. Second, our results are higher than CQD on almost all metrics, like _Avg_p_ of ENeSy is __24.5__ and __29.4__ on FB15K-237 and NELL-995, respectively. While the results of CQD are __21.7__ and __28.4__. Finally, we __add__ CQD as a baseline in Table2 in the new version.\
> We didn't discuss GNN-QE[1] since it is a concurrent work with ours which was publicly available on arxiv on __May 16th__, only three days before our submission. We would like to add it as a __related work__ in the new version. Actually, compared to GNN-QE, ENeSy gets better results on NELL-995 with _Avg_p_  and _Avg_n_ as __29.4__ and __9.8__, respectively. GNN-QE gets results which are __28.9__ and __9.7__. What’s more, the key point of ENeSy is to explore how to enable neural and symbolic reasoning to support each other, while GNN-QE concentrates on interpretation of intermediate variables, which ENeSy can also provide with symbolic representation.
> - For Weakness4: We regard FuzzQE, which could be trained with only 1p query, as the main baseline. Thus we follow the experiments in FuzzQE[4] reporting results on FB15K-237 and NELL-995 which are trained with all queries and only 1p query. For the completeness of the table and the space limitation, we didn’t report the results on FB15K.
>
> For Question:
> - For Question3: We didn’t manually check the queries since they are generated randomly by a script. Meanwhile, the answer number of a query can be large like hundreds. For demonstration purposes, we select a query from NELL without too many answers and the __case study__ is presented in the appendix.
>
> For Limitation:\
> The improvement can be proved with Figure4(a). With our framework, the accuracy of embedding models improves a lot. The more complicated the query is, the more the results improve. Meanwhile, the experiment based on TransE can prove this, too. (Please refer to the reply to Weakness2).
>
> [1] Neural-Symbolic Models for Logical Queries on Knowledge Graphs. ICML 2022\
> [2] Complex query answering with neural link predictors. ICLR 2021\
> [3] Fuzzy Logic based Logical Query Answering on Knowledge Graphs. AAAI 2022

---

> ### Author Response · Authors · 2022-08-07
> **We welcome any questions about our response.**
>
> Dear reviewer, we appreciate any suggestions from you, and we welcome any more questions about our response. Looking forward to hearing from you.

---

### Official Review · Reviewer_QWSg · 2022-07-11

**Rating:** 7
**Confidence:** 2
**Soundness:** 3 good
**Presentation:** 4 excellent
**Contribution:** 3 good

**Summary:**

This paper proposes a neural symbolic entangled framework for answering complex logical queries over knowledge graph. The framework firstly represents entities and relations via a neural projection and a symbolic projection. Then the symbolic based logical operator is defined to combine the neural and symbolic projection. Results show the proposed framework outperforms baselines on complex query answering benchmarks.

**Questions:**

N/A

**Limitations:**

Yes

**Strengths And Weaknesses:**

Pros:
-	The proposed neural symbolic entangled framework is novel, which combines neural representation and symbolic representation of entities and relations.
-	This paper is well written and easy to follow


Cons:
-	The proposed approach does not fully outperform the strongest baseline FuzzQE on benchmark datasets

---

> ### Author Response · Authors · 2022-08-02
> **Response to Reviewer QWSg**
>
> We gratefully thanks for the precious time you spent and your positive feedback.

---

### Meta-Review · Area_Chair_pNRb · 2022-08-28

**Recommendation:** Accept
**Confidence:** Less certain

**Metareview:**

This paper proposes a neural-symbolic approach (ENeSy) for answering FOL-based queries over KGs. The main promise of the proposed approach lies in its ability to alleviate the problem of cascading error as it existed in previous approaches and also the incompleteness of KGs. The experiments are performed on two datasets (FB15K-237 and NELL-995) which established the merits of the proposed ENeSy over the baselines.

The reviewers are generally happy with the clarity and substance of the work. There are some suggested baselines in "3t6R"'s review which (to my understanding) are addressed in the authors' response.   The experiments on computation speed (in response to "7Bea") are also informative.  I hope all of these will be reflected in their revised draft.  Lastly, I hope the authors release the scripts necessary for reproducing their experiments.

**Award:**

No

---

### Decision · Program_Chairs · 2022-09-14

Accept